# Solving Constrained Optimization Problems as ODE-based Models Using Reinforcement Learning

**Han Meng** *hmeng@wm.edu*
*College of William and Mary*
**Xinsong Feng** *xfeng06@wm.edu*
*College of William and Mary*
**Yang Li** *yli102@wm.edu*
*College of William and Mary*
**Chenan Wang** *cwang33@wm.edu*
*College of William and Mary*
**Kishansingh Rajput** *kishan@jlab.org*
*Thomas Jefferson National Accelerator Facility*
**Malachi Schram** *malachi.schram@pnnl.gov*
*Pacific Northwest National Laboratory*
**Haipeng Chen** *hchen23@wm.edu*
*College of William and Mary*

**Reviewed on OpenReview:** *https://openreview.net/forum?id=QWOZX4zRC2*

## Abstract

Previous learning-to-optimize (L2O) methods on constrained optimization problems often treat neural networks as initializers that generate approximate solutions requiring substantial post-hoc refinements. This approach overlooks a key insight: Solving complex optimization problems often requires iterative refinement of candidate solutions, a process naturally aligned with the Markov Decision Process (MDP) and reinforcement learning (RL) framework. We show that within the MDP framework, RL and Ordinary Differential Equation (ODE)-based generative models (e.g., diffusion, flow matching) are formally equivalent, unifying them as trainable optimizers. Building on our unified perspective, we propose to train a flow-matching model within an RL paradigm as a learnable refinement mechanism, thereby incorporating constraint satisfaction directly into the optimization process. To further enhance feasibility, we introduce a minimal correction step that adjusts solutions to ensure constraint compliance. Empirical results demonstrate that our approach achieves state-of-the-art performance across a range of constrained optimization tasks, yielding improvements in inference speed, solution quality, and feasibility over prior baselines. Our code is available on GitHub.

## 1 Introduction

Constrained optimization covers a broad spectrum of applications in science and engineering, such as power systems (Fioretto et al., 2020), portfolio selection (Das et al., 2023), robotics trajectory planning (Li et al., 2025a), and real-time resource allocation (Różańska & Horn, 2024). Classical solvers, convex optimization methods for tractable problems, and heuristic solvers for non-convex ones, are principled but often too slow for real-time scenarios (Dong et al., 2020). Learning to optimize (L2O) offers a promising direction for accelerating inference (Koziel & Leifsson, 2013).

Recent progress in L2O can broadly be categorized into two lines. The first typically uses a neural network to propose partial solutions that are then completed by enforcing equality constraints, followed by iterative refinement. Advances in this line largely stem from borrowing mechanisms from classical optimiza-

tion, such as primal-dual methods and augmented Lagrangian updates, to design gradient-based refinement steps (Agrawal et al., 2019; Amos & Kolter, 2017; Donti et al., 2021; Park & Van Hentenryck, 2023; Tanneau & Van Hentenryck, 2024; Klamkin et al., 2024). These algorithms mostly focus on limited settings and remain suboptimal, as they heavily rely on iterative post-processing to refine solutions. The second line explores constrained sampling with diffusion-based generative models, with applications to trajectory optimization and related domains (Janner et al., 2022; Pan et al., 2024; Li et al., 2025a; Zhang et al., 2025). Yet, it remains a widely recognized challenge to enforce strict adherence to complex, high-dimensional equality and inequality constraints (Liang & Chen, 2024; Li et al., 2025b; Ding et al., 2025).

Inspired by iterative optimization algorithms like Interior Point methods (Potra & Wright, 2000), our key insight is: *Solving complex optimization problems often requires iterative refinement of candidate solutions.* Following this insight, we formalize constrained optimization as a generic MDP, where the state represents the current candidate solution, the action corresponds to a refinement step, and the reward encodes progress toward feasibility and optimality. Solving the optimization problem thus amounts to learning an reinforcement learning (RL) policy that incrementally improves solutions until they reach the feasible region and achieve near-optimal objective values.

ODE-based generative models, including diffusion and flow matching (Lipman et al., 2023), instantiate this idea naturally. Diffusion models iteratively denoise a random initialization, progressively steering samples toward the target distribution (Ho et al., 2020; Song et al., 2021; Sohl-Dickstein et al., 2015). Viewing each denoising step as an action and the data distribution as the implicit MDP target, diffusion can be interpreted as training policies to update candidates from arbitrary distributions into the target.

Despite advances in ODE-based generative models, applying them to constrained problems remains difficult. Equality constraints shrink the feasible set to a narrow manifold, making optimization ill-conditioned, while constrained losses often fail under tight tolerances as solutions oscillate near boundaries. Post-processing methods, such as completion (Donti et al., 2021) or mapping (Li et al., 2025b), enforce feasibility but disrupt optimization: completion-based approaches (Agrawal et al., 2019) often suffer from instability, whereas mapping-based ones (Christopher et al., 2024) can degrade objectives and add overhead. By reshaping critical feedback signals, these methods ultimately limit RL policy training and overall performance.

To address these challenges, we revisit the nature of constrained optimization and formally establish the first RL-based framework for iteratively solving constrained optimization. We first provide a unified view of RL and ODE-based generative models through the lens of iterative solution dynamics. Building on this, we propose the constrained Markov flow optimizer (CMFO), a flow-matching-based trainable optimizer tailored to constrained optimization and trained within the RL paradigm. In addition, we analyze the failure modes of existing methods for enforcing constrained generation in ODE-based models and introduce the *relaxComplete* operator, which preserves the full reward signal while providing necessary assistance in constraint enforcement.

We conduct extensive experiments to evaluate the performance of CMFO across a range of constrained optimization tasks, including convex quadratic programs (QPs), non-convex quadratic programs with sine regularization (QPSR), and the practical AC optimal power flow (ACOPF) problem. Compared with traditional one-step L2O methods, ODE-based methods are generally superior in generating solutions with better objective values and faster inference speed. Among ODE-based methods, CMFO achieves superior optimality with the best feasibility across high-dimensional convex, non-convex, and power-system benchmarks.

## 2 Related Work

We briefly introduce existing work in L2O. A more detailed discussion is in Appendix A.

**One-step L2O methods.** This line of work seeks to accelerate the solution of complex problems, often under constraints. Early surrogate models directly map problem instances to solutions (Koziel & Leifsson, 2013) but struggle with feasibility and optimality. Later approaches use neural predictors to warm-start classical solvers, improving convergence and robustness (Baker, 2019; Dong et al., 2020). More recently, end-to-end pipelines couple predictions with explicit constraint handling and refinement, such as DC3 (Donti et al., 2021), which combines completion and gradient-based corrections. Extensions improve these steps

with dual-inspired updates (Park & Van Hentenryck, 2023; Tanneau & Van Hentenryck, 2024; Klamkin et al., 2024). Our approach follows this paradigm but differs in framing refinement as an RL problem, where a neural policy proposes iterative updates and is supported by constraint-aware operators.

**Diffusion models-based L2O methods.** A closely related line of work in L2O is based on diffusion models (Sohl-Dickstein et al., 2015; Song et al., 2021; Ho et al., 2020; Lipman et al., 2023), which optimizes implicitly toward objectives defined by the data distribution and thus can be viewed as a model-free setting. Subsequent variants introduce model-based formulations (Pan et al., 2024) and improve sampling quality and efficiency (Berner et al., 2024; Richter & Berner, 2024; Chen et al., 2025). For constrained settings, methods embed constraints in the loss (Pan et al., 2024), warm-start classical solvers (Li et al., 2025a), adapt dual methods (Zhang et al., 2025), or enforce feasibility through projections (Christopher et al., 2024) and gauge mappings (Li et al., 2025b). While effective in some perspectives, these approaches fail to deal with the challenging trade-offs among feasibility, optimality, and inference time in constrained optimization problems.

**RL for optimization.** RL has been extensively applied to *discrete, combinatorial optimization* (Mazyavkina et al., 2021), where the discrete nature of the search space often renders exact solvers computationally infeasible. Initial successes have been achieved in solving the traveling salesman problem (TSP) (Bello et al., 2017), and the vehicle routing problem (VRP) (Nazari et al., 2018). These initial successes spur a wave of research on architectural innovations that integrate RL with graph representation learning (Khalil et al., 2017; Chen & Tian, 2019; Lu et al., 2020; Kwon et al., 2020; Chen et al., 2021; Hottung et al., 2022; Feng et al., 2025; Li et al., 2025c), shifting the focus from direct policy learning toward hybrid approaches that combine RL with explicit search or refinement mechanisms. By contrast, RL is only sparingly applied to *continuous optimization*. A few studies explore its use for general continuous optimization (Li & Malik, 2017a;b; Chen et al., 2022) or in domain-specific contexts (Xian et al., 2025), but none address *constrained* continuous optimization, which is the focus of our work.

**Intersection of RL and ODE-based generative model** Some related works (Wang et al., 2023; Ma et al., 2025; McAllister et al., 2025; Pfrommer et al., 2025; Park et al., 2025) have explored the intersection between reinforcement learning and ODE-based models, which mostly utilize generative models as an expressive policy.

# 3 Constrained Optimization Via the Lens of Iterative Solution Dynamics

Optimization is usually expressed in static form-minimizing an objective subject to constraints. In practice, however, solvers act *iteratively*, refining candidate solutions under structural feedback. This dynamic view motivates our key insight: *iterative constrained optimization is naturally an RL problem.* Unlike classical solvers with hand-crafted updates, RL treats the update rule as a policy that can be trained and optimized. Following this insight, we present the first systematic formulation of constrained optimization as an MDP, casting iterative refinement as a sequential decision process. As a natural extension, we further connect this RL view to ODE-based generative models, whose dynamics correspond to continuous-time policies. Under this unified lens, RL and generative modeling are both reinterpreted as instances of *trainable optimizers*, bridging two previously disjoint lines of research under a common foundation.

## 3.1 The Constrained Optimization Problem

Consider a parametric family of constrained optimization problems, each defined by an instance $X \in \mathcal{X}$. For a given instance $X$, let the objective be $f_X : \mathbb{R}^d \to \mathbb{R}$ with inequality and equality constraints $g_X : \mathbb{R}^d \to \mathbb{R}^{m_{\text{ineq}}}$ and $h_X : \mathbb{R}^d \to \mathbb{R}^{m_{\text{eq}}}$. Formally, we have

$$\min_{y \in \mathbb{R}^d} \ f_X(y) \quad \text{s.t.} \quad g_X(y) \preceq \mathbf{0}, \quad h_X(y) = \mathbf{0}. \tag{1}$$

The structure of $f_X, g_X, h_X$ and the optimal solution $y^*(X)$ depend on the instance $X$, and may be nonlinear and nonconvex. This standard static formulation will serve as the basis for our dynamic reinterpretation.

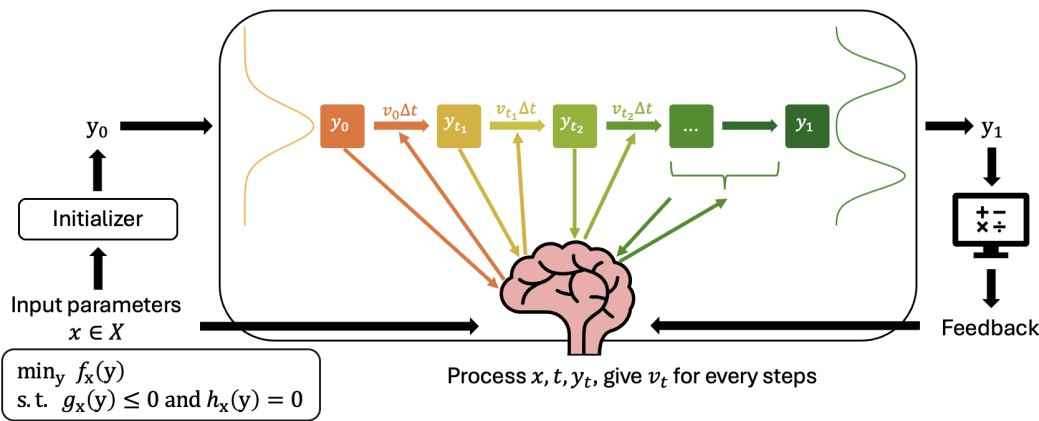

Figure 1: Flow-matching models solve the optimization problems as MDPs.

## 3.2 Constrained Optimization as an MDP

We now formalize constrained optimization as an episodic MDP, making its iterative nature explicit.

For each problem instance $X$, we define $\mathcal{M}_X = \langle \mathcal{S}, \mathcal{A}, P, R, \gamma \rangle$. At iteration $t$, the state is $s_t = (X, y_t)$, consisting of the problem instance $X$ and the current solution candidate $y_t$. An action $a_t = \Delta y_t$ corresponds to an update step that deterministically produces $y_{t+1} = y_t + \Delta y_t$, with optional stochasticity introduced by noise. The reward $r_t = M_X(y_t) - M_X(y_{t+1})$ measures progress via the decrease of a merit function:

$$M_X(y) = f_X(y) + \alpha \|h_X(y)\|_2 + \beta \|[g_X(y)]_+\|_2. \tag{2}$$

Here $\| \cdot \|_2$ is the L-2 norm, $[u]_+ = \max(u, 0)$ is applied element-wise, and $(\alpha, \beta)$ are penalty weights chosen sufficiently large to approximate hard constraints.

Since only the terminal solution matters, we adopt the undiscounted setting $\gamma = 1$. The cumulative reward then telescopes to

$$\begin{aligned} J(\pi) &= \sum_{t=0}^{T-1} \left( M_X(y_t) - M_X(y_{t+1}) \right) \\ &= M_X(y_0) - M_X(y_T). \end{aligned} \tag{3}$$

Given $M_X(y_0)$ is regarded as constant, maximizing return is therefore equivalent to minimizing the terminal merit $M_X(y_T)$, showing that the MDP formulation is not merely an analogy but a faithful restatement of the original constrained optimization objective.

## 3.3 ODE-based Generative Models as MDPs

We now turn to ODE-based generative models, which can be described under a unified continuous-time framework

$$dx_t = f(x_t, t)dt, \tag{4}$$

where the learnable vector field $f$ prescribes how the sample $x_t$ should evolve over time. The generative process is simply the numerical integration of this ODE, starting from a prior (noise) distribution $p_0$ and converging toward a target distribution $p_1$. Different choices of $f$ recover well-known models.

**Flow matching.** In this case, the ODE dynamics $dx_t/dt = f(x_t, t)$ are parameterized directly by a learnable velocity field $v_\theta$. Supervision is obtained by constructing a simple interpolation between a noise sample $x_0 \sim p_0$ and a data sample $x_1 \sim p_1$, $\bar{x}_t = (1 - \alpha(t))x_0 + \alpha(t)x_1$, whose analytic velocity is $v^\star(\bar{x}_t, t) = \dot{\alpha}(t)(x_1 - x_0)$. The training objective is then to regress $v_\theta$ toward this oracle velocity along the path,

$$\mathcal{L}_{\mathrm{FM}}(\theta) = \mathbb{E}_{t, x_0, x_1} \left[ \|v_\theta(\bar{x}_t, t) - v^\star(\bar{x}_t, t)\|^2 \right]. \tag{5}$$

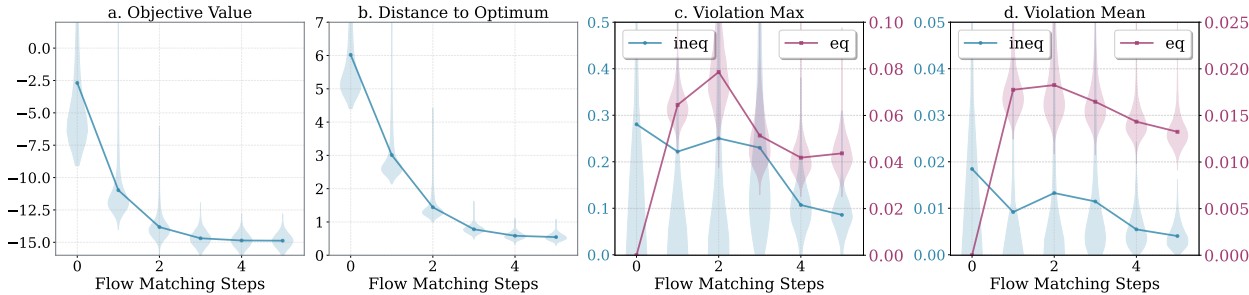

Figure 2: Performance metrics across steps for VANILLA. Smaller values are better for all metrics.

**Diffusion models.** Diffusion starts from a stochastic forward process that perturbs data into noise. Song et al. (2021) showed that the reverse dynamics admit a Probability Flow ODE, $\frac{dx_t}{dt} = \frac{x_t - \mathbb{E}[x_0|x_t]}{t}$, where $\mathbb{E}[x_0 \mid x_t]$ is the denoiser. In practice, a neural network $D_\phi(x_t, t)$ is trained with the denoising score-matching loss $\mathbb{E}\|x_0 - D_\phi(x_t, t)\|^2$ to approximate this conditional expectation, yielding the empirical PF ODE $\frac{dx_t}{dt} = \frac{x_t - D_\phi(x_t, t)}{t}$.

Once the vector field $f$ is learned, the generative process reduces to numerically integrating the dynamics. In practice, this is implemented as discrete updates such as

$$x_{k+1} = x_k + \Delta t \cdot f(x_k, t_k), \tag{6}$$

so that the generation proceeds step by step from an initial noise sample toward the data distribution.

This iterative structure makes the MDP connection immediate: the state is $(x_t, t)$, the action is given by $f(x_t, t)$, the transition is given by the integration scheme, and the reward reflects how close the trajectory approaches the target distribution. Viewed this way, ODE-based generative models are just continuous-time policies trained by imitation, fitting neatly into the RL framework introduced in Section 3.2. A detailed discussion is in Appendix B.1.

## 4 Constrained Markov Flow Optimizer (CMFO)

As discussed in Section 3.3, diffusion and flow-matching models can both be cast as continuous-time policies under the RL framework. Diffusion relies on Gaussian priors and noise-driven trajectories, which often produce long, unstable paths and require careful schedule design. In contrast, flow matching parameterizes deterministic ODE dynamics that directly transport an initializer distribution to the target, naturally mirroring constrained optimization where candidates are refined toward feasibility and optimality. This direct and stable formulation makes flow matching a more natural and efficient backbone for our optimizer, with exploration complemented by metric function rather than stochastic sampling.

We now introduce CMFO, a learnable optimizer composed of a flow-matching refinement module (Figure 1) and a stabilized constraint-handling operator. The model takes warm-start solutions from a lightweight neural network–a standard practice–and iteratively refines them to improve both optimality and feasibility.

### 4.1 Challenges of a Vanilla Implementation

At first glance, the alignment between flow matching and the MDP view suggests a straightforward route to constrained optimization, i.e., to enforce constraints by adding penalty terms to the loss function, as shown in Equation 2. With the established equivalence between the formulations of flow-matching and RL (Section 3.3), this loss can be seen as a Q-value-style objective that drives iterative refinement. We call this baseline the VANILLA method.

Figure 2 reports a preliminary study on its performance across four metrics, where the x-axis is the flow matching steps, and the y-axis represents the metric value that is described by the respective sub-caption.

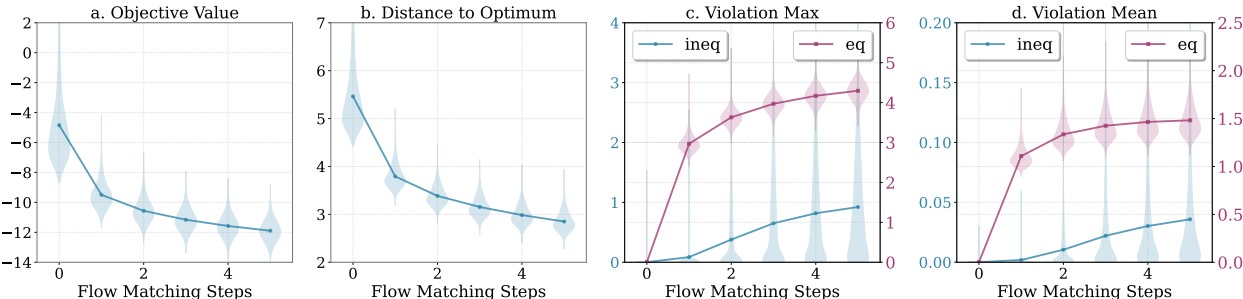

Figure 3: Performance metrics across steps for COMPLETION (before completion operator). Smaller values are better for all metrics.

We can see that the objective value (Figure 2a), the distance to the optimum (Figure 2b), and the maximum and mean numbers of violations of inequality constraints (blue curve in Figure 2c,d) all consistently show a steady trend of decrease. For *equality constraints* (red curve in Figure 2c,d), however, violations initially increase, since the initialization model enforces feasibility through completion, but later decrease as refinement proceeds. Nevertheless, the final violations remain non-negligible. In fact, our empirical results (Table 1) show that the VANILLA method ultimately achieves 0% equality feasibility.

In summary, while the VANILLA method improves objective values and reduces inequality violations, it fails to enforce equality constraints reliably. This limitation stems from the ill-conditioned optimization landscape near the feasible region (e.g., functions like $|x - a|$), where gradients are unstable or uninformative, thereby hindering precise convergence.

## 4.2 RelaxCompletion on Equality Constraints

**Completion.** Following (Donti et al., 2021), this technique enforces equality constraints by optimizing only over a reduced set of free variables. Given $y \in \mathbb{R}^n$, let $y_{\text{partial}}$ denote $m$ free entries, while the remaining $(n - m)$ entries are recovered via a mapping $\varphi(y_{\text{partial}})$ such that

$$h_x\left(\left[\, y_{\text{partial}} \,\|\, \varphi(y_{\text{partial}}) \,\right]\right) = 0, \tag{7}$$

where $\|$ denotes concatenation and $\varphi(\cdot)$ may be either explicit or implicit (e.g., obtained via Newton's method). This reformulation constrains optimization to the equality-constraint manifold. By the implicit function theorem, gradients can be backpropagated through $\varphi(\cdot)$ regardless of its differentiability (Donti et al., 2021; Amos & Kolter, 2017).

**Zero gradient propagation for enforced constraints.** While equality completion can strictly enforce equality constraints, its key limitation is that it removes the influence of equality constraints during training, delegating enforcement entirely to the operator. Because solutions are projected directly into the equality-feasible region, the corresponding loss terms, and thus their gradients, remain identically zero.

Figure 3 summarizes the refinement dynamics of the flow matching model in the COMPLETION method. Compared with the VANILLA baseline (Figure 2), the most notable difference is that violations of equality constraints consistently increase during refinement to large values (Figure 3c,d), indicating complete reliance on the operator for enforcement. This also suggests that the adjustments introduced by completion to candidate solutions can be substantial, which in turn degrades other performance metrics, such as the objective value (Figure 3a), distance to the optimum (Figure 3b), and inequality constraints (Figure 3c,d), relative to the VANILLA method. These findings highlight the importance of feedback from equality constraints in guiding refinement, motivating new constraint-handling techniques.

**RelaxCompletion.** To overcome the limitations of strict completion, we propose *relaxCompletion* (*relaxComp*), a soft operator that retains the learning signal of equality constraints while guiding solutions toward feasibility. Given a candidate solution $y \in \mathbb{R}^n$, let $\mathsf{C}(y) = \left[\, y_{\text{partial}} \,\|\, \varphi(y_{\text{partial}}) \,\right]$ denote the solution obtained

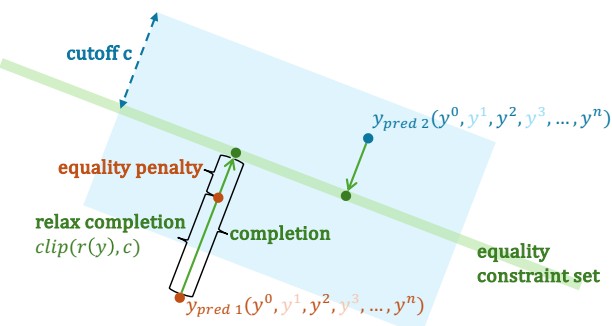

Figure 4: **Illustration of relaxCompletion.** The blue region denotes the cutoff radius $c$ that limits the magnitude of the correction applied to model outputs. For $\mathbf{y}_{\text{pred 1}}$, which is far from the equality-constrained set, the allowable correction (bounded by $c$) is insufficient to reach the set, leaving a non-zero equality residual that provides useful gradients. In contrast, $\mathbf{y}_{\text{pred 2}}$ is close enough that the clipped correction reaches the set and fully satisfies the equality constraints.

by COMPLETION and define the residual

$$r(y) = \mathsf{C}(y) - y, \tag{8}$$

which captures the update introduced by enforcing feasibility. Instead of applying $r(y)$ in full, a bounded correction is conducted:

$$y^+ = y + \text{clip}\left(r(y),\, c\right), \tag{9}$$

where $c > 0$ is a cutoff parameter and $\text{clip}(\cdot, c)$ applies elementwise clipping in $[-c, c]$.

Figure 4 illustrates how *relaxComp* helps during training and inference. First, it provides full guidance for refinement, guiding prediction toward improved objective values and feasibility of both equality and inequality constraints, alleviating instability from post-processing, an insight that coincides with PPO (Schulman et al., 2017). Second, it allows direct loss backpropagation to the model, avoiding expensive Jacobian computations and scalability issues in high-dimensional settings. Finally, when the solution is already close to the optimum and the feasible region, where the vanilla approach reaches its limit, *relaxComp* stands out and applies only a minimal corrective step to ensure feasibility. Since this adjustment is small, its effect on the objective value and inequality constraints is often negligible.

### 4.3 Training and Inference of CMFO

---
**Algorithm 1** Training Algorithm for Constrained Markov Flow Optimizer
---
**Require:** $\mathcal{D}_{train} = \{(X, Y)\}$; initialization model Init; flow-matching based OPT with velocity predictor $vm$; constrained loss $\ell_q(\cdot)$, behavior cloning loss $\ell_{bc}(\cdot, \cdot, \cdot)$; weights $w_{\text{bc}}, w_{\text{q}}$

1: **Initialize** parameters of Init and $vm$
2: **for** epoch $= 1, 2, \ldots, N_{\text{epoch}}$ **do**
3:     **for** each minibatch $(X, Y) \sim \mathcal{D}$ **do**
4:         # train initialization model
5:         $\hat{y} \leftarrow \text{Complete}(\text{Init}(X))$
6:         $\mathcal{L}_{init} \leftarrow \ell_q(\hat{y})$ and update Init using $\mathcal{L}_{init}$
7:         # train flow-matching based optimizer
8:         $y_{\text{cand}} \leftarrow \text{OPT}(X, \hat{y})$
9:         $y \leftarrow \text{RELAXCOMPLETE}(y_{\text{cand}})$
10:        $\mathcal{L}_{\text{bc}} \leftarrow \ell_{bc}(X, \hat{y}, Y); \quad \mathcal{L}_{\text{q}} \leftarrow \ell_q(y)$
11:       $\mathcal{L}_{\text{total}} \leftarrow w_{\text{bc}}\mathcal{L}_{\text{bc}} + w_{\text{q}}\mathcal{L}_{\text{q}}$
12:       Update $vm$ using $\mathcal{L}_{\text{total}}$
13:       Update Init using $\mathcal{L}_{\text{total}}$ Optionally

---

**Training.** As discussed in Section 3.3, ODE-based generative models can be viewed as fixed-step RL methods, where predefined templates construct trajectories in the absence of explicit data. Section 3.2 further shows that constrained optimization can be cast as a deterministic, undiscounted MDP, with returns directly derived from the optimization objective. In practice, such optimization is naturally finite-horizon due to time and computational limits. Accordingly, CMFO training follows the RL paradigm, combining behavior cloning and Q value losses to guide the optimization process (Algorithm 1).

As discussed in Section 3.2, maximizing the return is equivalent to minimizing the merit function defined in Equation 2 in our problem setting. Thus, we defined the Q value loss, $\ell_q$, to minimize Equation 2.

$$l_q(X, y) = f(x, y) + \alpha \|h(X, y)\|_2 + \beta \|[g(X, y)]_+\|_2. \tag{10}$$

where $X$ is problem instance, $y$ is final predicted solution. While the behavior cloning loss, $\ell_{bc}$, is derived from the standard flow-matching loss in Equation 5 (equivalence has been discussed in Section 3.3):

$$\ell_{bc}(X, \hat{y}, Y) = \|v_\theta(X, y_t, t) - (Y - \hat{y})\|^2, \tag{11}$$

where $X$ is problem instance, $\hat{y}$ is initial solution, $Y$ is ground-truth solution, and $y_t = (1 - t)\hat{y} + tY$, $t \sim \mathcal{U}(0, 1)$.

The overall training objective for updating the velocity (and optionally the initialization) model is given by

$$\mathcal{L}_{\text{total}} = w_{\text{bc}} \ell_{\text{bc}} + w_{\text{q}} \ell_{\text{q}}, \tag{12}$$

where $w_{\text{bc}}$ and $w_{\text{q}}$ are weights controlling the relative contributions of behavior cloning and Q value optimization. Analogous to RL, behavior cloning encourages the model to imitate high-quality trajectories, while the Q value loss promotes exploration of the solution space and guides the model toward better solutions. In practice, we adopt a two-stage training schedule: during the first stage, both terms are active, with behavior cloning in a dominating role to stabilize learning. In the second stage, training relies solely on Q value optimization to further refine the model.

---

**Algorithm 2** RELAXCOMPLETION

---

**Require:** candidate $y \in \mathbb{R}^n$, completion operator $\mathsf{C}(\cdot)$, cutoff $c > 0$
1: **function** RELAXCOMPLETION($y, \mathsf{C}, c$)
2:     $r \leftarrow \mathsf{C}(y) - y$
3:     $r_{\text{clip}} \leftarrow \text{clip}(r, -c, c)$
4:     **return** $y^+ \leftarrow y + r_{\text{clip}}$

---

**Inference.** Given an instance $X$, inference proceeds in three stages:

*(1) Warm start.* Predict an initial guess $\tilde{y} = \text{Init}(X)$ using initialization model and map it into the equality manifold using equality completion (Equation 7).

*(2) Flow-matching refinement.* Starting from $\hat{y}$, run the learned optimizer to produce a candidate $y_{\text{cand}}$.

*(3) Soft feasibility enforcement.* Apply the *relaxComp* operator to $y_{\text{cand}}$ to obtain the final iterate $y$ according to Algorithm 2.

To further improve feasibility and optimality, one may batch-sample multiple solutions and select the best feasible one. This increases computational cost, but typically only increases negligible wall-clock time when executed in parallel.

## 5 Experiment

### 5.1 Experimental Setup

**Tasks.** Following DC3 (Donti et al., 2021), we evaluate on three representative problems: (i) convex quadratic programs (QPs), (ii) non-convex quadratic programs with sine regularization (QPSR), and (iii)

Table 1: Comparison of different methods on QPs.

| Metric | Baselines | | | CMFO | | |
|---|---|---|---|---|---|---|
| | **Optimizer** | **DC3** | **B-Projection** | **Vanilla** | **+Completion** | **+RelaxComp (ours)** |
| Obj. value ↓ | **-15.05 (0.00)** | -13.45 (0.03) | -4.38 (0.15) | -14.83 (0.04) | -14.54 (0.09) | -14.17 (0.08) |
| Max eq. viol. ↓ | 0.00 (0.00) | 0.00 (0.00) | 0.00 (0.00) | 0.03 (0.01) | 0.00 (0.00) | 0.00 (0.00) |
| Mean eq. viol. ↓ | 0.00 (0.00) | 0.00 (0.00) | 0.00 (0.00) | 0.01 (0.00) | 0.00 (0.00) | 0.00 (0.00) |
| Max ineq. viol. ↓ | 0.00 (0.00) | 0.00 (0.00) | 0.19 (0.00) | 0.01 (0.00) | 0.07 (0.06) | 0.02 (0.03) |
| Mean ineq. viol. ↓ | 0.00 (0.00) | 0.00 (0.00) | 0.00 (0.00) | 0.00 (0.00) | 0.00 (0.00) | 0.00 (0.00) |
| Eq. fea. rate ↑ | 1.00 (0.00) | 1.00 (0.00) | 1.00 (0.00) | 0.00 (0.00) | 1.00 (0.00) | 1.00 (0.00) |
| Ineq. fea. rate ↑ | 1.00 (0.00) | 1.00 (0.00) | 0.63 (0.01) | 0.62 (0.22) | 0.51 (0.24) | 0.75 (0.35) |
| Fea. rate ↑ | **1.00 (0.00)** | **1.00 (0.00)** | 0.63 (0.01) | 0.00 (0.00) | 0.51 (0.24) | 0.75 (0.35) |
| Time (s) ↓ | 1.692 (0.013) | 0.009 (0.000) | 0.017 (0.001) | **0.002 (0.000)** | 0.008 (0.000) | 0.008 (0.000) |

Table 2: Comparison across methods on QPSR.

| Metric | Baselines | | | CMFO | | |
|---|---|---|---|---|---|---|
| | **Optimizer** | **DC3** | **B-Projection** | **Vanilla** | **+Completion** | **+RelaxComp (ours)** |
| Obj. value ↓ | **-11.59 (0.00)** | -10.70 (0.02) | -7.56 (0.15) | -11.47 (0.00) | -11.30 (0.10) | -11.29 (0.04) |
| Max eq. viol. ↓ | 0.00 (0.00) | 0.00 (0.00) | 0.00 (0.00) | 0.02 (0.00) | 0.00 (0.00) | 0.00 (0.00) |
| Mean eq. viol. ↓ | 0.00 (0.00) | 0.00 (0.00) | 0.00 (0.00) | 0.01 (0.00) | 0.00 (0.00) | 0.00 (0.00) |
| Max ineq. viol. ↓ | 0.00 (0.00) | 0.00 (0.00) | 0.01 (0.00) | 0.01 (0.00) | 0.09 (0.12) | 0.01 (0.01) |
| Mean ineq. viol. ↓ | 0.00 (0.00) | 0.00 (0.00) | 0.00 (0.00) | 0.00 (0.00) | 0.00 (0.00) | 0.00 (0.00) |
| Eq. fea. rate ↑ | 1.00 (0.00) | 1.00 (0.00) | 1.00 (0.00) | 0.00 (0.00) | 1.00 (0.00) | 1.00 (0.00) |
| Ineq. fea. rate ↑ | 1.00 (0.00) | 1.00 (0.00) | 0.99 (0.00) | 0.82 (0.06) | 0.60 (0.40) | 0.91 (0.04) |
| Fea. rate ↑ | **1.00 (0.00)** | **1.00 (0.00)** | 0.99 (0.00) | 0.00 (0.00) | 0.60 (0.40) | 0.91 (0.04) |
| Time (s) ↓ | 0.192 (0.010) | 0.006 (0.000) | 0.024 (0.011) | **0.002 (0.000)** | 0.008 (0.000) | 0.008 (0.000) |

the real-world AC optimal power flow (ACOPF) problem. For the two synthetic datasets, QPs and QPSR, we set the solution dimension to 100, with both the number of equality and inequality constraints fixed at 50. Details of these three problems can be found in Appendix C.

**Metrics.** We assess the performance of different methods across three dimensions:

- *Optimality*: We use the objective value $f_x(y)$ (*obj. value*) to quantify the optimality of a final solution.
- *Feasibility*: We provide fine-grained metrics to evaluate the feasibility violations by the final solutions. *Max/Mean ineq/eq viol.* are the maximum and mean number of inequality or equality feasibility violations quantified by $\text{ReLU}(g_X(y))$ or $|h_X(y)|$, respectively. *Ineq/Eq fea. rate* are the ratios of samples that are considered ineq/eq feasible with all violations $< \epsilon$. *Fea. rate* is the portion of samples that satisfy both the inequality and equality feasibility. The threshold $\epsilon$ is set to be 0.0001.
- *Efficiency*: We use the inference time to characterize the efficiency. Shorter time means higher efficiency.

**Baselines.** We adopt three baselines from prior work.

- OPTIMIZER: Classical solvers including qpth (Stellato et al., 2020) for QPs, IPOPT (Wächter & Biegler, 2006) for QPSR, and PYPOWER (Zimmerman et al., 1997) for ACOPF.
- DC3: Generates a partial candidate solution using a neural network, completes it via equality completion, and refines it with gradient-based corrections (Donti et al., 2021).
- B-PROJECTION: Uses interior-point prediction plus bisection to project NN outputs onto general constraint sets (Liang & Chen, 2025).

We also craft two baselines that ablate our method.

- VANILLA: A flow-matching-based optimizer trained with $\mathcal{L}_{\text{total}}$. It removes *relaxComp* in our method.
- COMPLETION: Replace *relaxComp* in our method with equality completion.

We use +RELAXCOMP to represent our method in the ablation. For a fair comparison, all flow-matching-based methods used an initialization model to generate solutions for timestep 0.

**Hyperparameters.** Unless otherwise noted, we set: $T = 5$ (timesteps of the flow-matching optimizer); $\alpha = \beta = 5$ in $\ell_q$; $c = 0.5$ in *relaxComp*; and $w_q = 0.001$. For ACOPF57, we set $w_{bc} = 0$. For QPs and QPSR,

Table 3: Comparison across methods on ACOPF.

| Metric | Baselines | | | CMFO | | |
|---|---|---|---|---|---|---|
| | **Optimizer** | **DC3** | **B-Projection** | **Vanilla** | **+Completion** | **+RelaxComp (ours)** |
| Obj value ↓ | 3.81(0.00) | 3.82 (0.00) | 3.90 (0.00) | **2.84 (0.58)** | 3.91 (0.07) | 3.82(0.00) |
| Max eq. viol. ↓ | 0.00 (0.00) | 0.00 (0.00) | 0.00 (0.00) | 0.35 (0.22) | 0.00 (0.00) | 0.00 (0.00) |
| Mean eq. viol. ↓ | 0.00 (0.00) | 0.00 (0.00) | 0.00 (0.00) | 0.06 (0.04) | 0.00 (0.00) | 0.00 (0.00) |
| Max ineq. viol. ↓ | 0.00 (0.00) | 0.00 (0.00) | 0.04 (0.00) | 0.00 (0.00) | 0.07 (0.05) | 0.01 (0.00) |
| Mean ineq. viol. ↓ | 0.00 (0.00) | 0.00 (0.00) | 0.00 (0.00) | 0.00 (0.00) | 0.00 (0.00) | 0.00 (0.00) |
| Eq. fea. rate ↑ | 1.00 (0.00) | 0.87 (0.01) | 1.00 (0.00) | 0.00 (0.00) | 1.00 (0.00) | 1.00 (0.00) |
| Ineq. fea. rate ↑ | 1.00 (0.00) | 0.62 (0.00) | 0.09 (0.01) | 1.00 (0.00) | 0.25 (0.27) | 0.54 (0.06) |
| Fea. rate ↑ | **1.00 (0.00)** | 0.58 (0.00) | 0.09 (0.01) | 0.00 (0.00) | 0.25 (0.27) | 0.54 (0.06) |
| Time (s) ↓ | 0.524 (0.004) | 0.090 (0.002) | 0.462 (0.012) | **0.003 (0.000)** | 0.042 (0.000) | 0.041 (0.001) |

$w_{bc}$ is initialized at 1 and decayed to 0 during 500-1000 epochs. All experiments are run on one NVIDIA A40 GPU for fair comparison.

Comparisons on the three datasets are presented in Tables 1, 2, and 3. Arrows indicate the desired trend of each metric (↑ for higher, ↓ for lower). Results are reported as mean with standard deviation (in brackets) over three runs. And we only sample once for every method.

## 5.2 Benchmark Results

We compare the three tasks separately.

**QPs** represent a simple optimization space with convex assumptions. In Table 1, we observe that RELAX-COMP strikes the best trade-off among optimality (obj. value), feasibility, and inference speed. Compared to the classical solver (qpth), ours is much more efficient, using only 0.4% inference time to achieve a similar objective value and feasibility. Using a similar run time to ours, DC3 and B-PROJECTION remain a large gap in the optimality.

In the task, ablating RELAXCOMP causes the solutions to violate constraints severely. None of the solutions by VANILLA satisfies the *equality constraints*. COMPLETE mitigates this issue for equality constraints but covers only 51% of inequality constraints. RELAXCOMP delivers the best overall feasibility, with only a mild cost in efficiency and optimality.

**QPSR** provides a harder test than QPs with a non-convex objective from sine regularization. RELAXCOMP still achieves a well-balanced performance, reaching the second-lowest objective value in 0.008 seconds, with 100% equality feasibility and 91%±4% inequality feasibility. By contrast, DC3 and B-PROJECTION sacrifice optimality substantially to satisfy the constraints.

The ablation results in the task are consistent with QPs. Worth mentioning, VANILLA finds the solution with an objective value even closer to the one by the classic optimizer. As a result, RELAXCOMP gets closer to the classic baseline in optimality as well.

**ACOPF** is a real-world optimization problem with inequality and highly non-convex equality constraints. Performance degradation consistently occurs with model-based methods, including DC3, B-PROJECTION, and CMFO variants. At a similar feasibility rate, our method finds the best solution in only 46% of the time used by DC3, showing superior efficiency under complex constraints. It also satisfies equality constraints better than DC3. Moreover, RELAXCOMP consistently outperforms B-PROJECTION in feasibility, optimality, and inference speed.

In the real-world task, the ablation study reiterates the value of our method. Fine-grained results show a non-trivial maximal feasibility violation, 0.35, by the VANILLA method. Simply adding equality completion can increase the overall feasibility rate to 25%. And RELAXCOMP can double the overall feasibility rate to 54%, which is a remarkable improvement.

**Overall**, we have these key findings from the three diverse tasks. (1) Our method balances the optimality, feasibility, and efficiency better than the traditional manually designed optimizers or the prior model-based

art. (2) RELAXCOMP achieves substantially better objective values than DC3, with a minimal decrease of optimality than that of other flow-matching–based methods. This difference can be attributed in part to the multi-objective nature of the constrained optimization loss and to the corrective adjustments. (3) Ablation studies also demonstrate the essence of *RelaxComp* in the CMFO framework in our method, without which constraints cannot be satisfied by the VANILLA methods or the COMPLETION.

Importantly, these findings highlight the promise of RL-augmented ODE-based generative models for tackling constrained optimization problems.

## 6 Conclusion

We presented CMFO, the first framework to solve constrained optimization by leveraging RL principles in conjunction with ODE-based generative models. Our formulation views constrained optimization as an MDP, where trajectories are generated following a predefined template inspired by flow-matching models, thereby enabling RL-style training without requiring explicit expert trajectories. To address the challenges of constraint enforcement, we introduced *relaxCompletion*, which retains full feedback signals while softly enforcing equality constraints. Across convex, non-convex, and real-world benchmarks, CMFO achieves a balanced combination of near-optimal objective values, strong feasibility, and fast inference, compared with DC3 and other flow-based baselines.

## Acknowledgements

This research was supported by the National Science Foundation under IIS-2348405. The authors also acknowledge the U.S. National Science Foundation for the computing resources support via the Advanced Cyberinfrastructure Coordination Ecosystem: Services & Support (ACCESS) program.

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
