# A    Related Work

**Learning to Optimize (L2O).** This line of work is motivated by the need to accelerate the solution of complex optimization problems, especially in the presence of constraints. Early approaches train surrogate models to map problem instances directly to solutions (Koziel & Leifsson, 2013), but such direct predictors often struggle with feasibility and optimality. Subsequent efforts use neural predictors to generate warm-starts for classical solvers, improving convergence speed and robustness (Baker, 2019; Dong et al., 2020). More recent work proposes end-to-end trainable pipelines that couple learned predictions with explicit constraint handling. For example, DC3 (Donti et al., 2021) predicts an approximate solution, applies a completion step to enforce equality constraints, and then performs gradient-based corrections toward feasibility and optimality. Building on this template, several methods improve the completion and correction steps with dual-inspired updates, including primal–dual schemes (Park & Van Hentenryck, 2023), augmented Lagrangian formulations (Tanneau & Van Hentenryck, 2024), and dual interior-point variants (Klamkin et al., 2024). Our approach follows this paradigm but differs in the refinement mechanism: rather than relying on closed-form, gradient-based corrections, we formulate refinement as an RL problem and employ a neural network to generate iterative updates, complemented by constraint-aware operators for handling feasibility.

**Diffusion models for optimization.** A closely related line of work applies diffusion-style generative models (Song et al., 2021; Ho et al., 2020; Lipman et al., 2023) to steer samples toward desired properties that can be cast as optimization objectives and/or constraints. Most formulations recast the task as sampling from an unnormalized target density. Early successes in the *model-free* setting, where only samples from the target distribution are available, motivate *model-based* variants in which the target is specified implicitly via objectives (Black et al., 2024), priors, or known properties. Several recent papers improve sampling *quality* or *efficiency* without explicit constraint handling (Berner et al., 2024; Richter & Berner, 2024; Chen et al., 2025).

Closer to constrained optimization, trajectory-optimization problems incorporate constraints into the target. Papers on this problem adopt different enforcement mechanisms. One line embeds objective/constraint terms directly in the training loss (Pan et al., 2024), but achieves low feasibility rates in practice. Another uses diffusion to *warm-start* classical solvers, delegating feasibility to the downstream optimizer (Li et al., 2025a), and works mainly for easy and low-dimensional trajectory tasks. Projection-based methods insert a projection onto the feasible set at every reverse step. PDM (Christopher et al., 2024) enforces hard constraints but is computationally expensive and can be numerically brittle. More recently, GFM enforces linear equality and convex inequality constraints via a bijective gauge mapping (Li et al., 2025b), offering elegant guarantees but with evidence primarily in low-dimensional tasks and without coupling to an explicit optimization objective at inference. In reinforcement-learning environments, (Zhang et al., 2025) adapts projected, primal–dual, and augmented-Lagrangian Langevin schemes within the reverse process to impose constraints directly; however, performance relies on expert-trajectory data. On the image generation task, (Fishman et al., 2023) proposes methods to handle inequality constraints.

Most prior art centers on diffusion models. Diffusion and flow matching are both powerful transport mechanisms; however, effectively and efficiently handling *both* equality and inequality constraints while maintaining progress on the optimization objective remains challenging. Existing approaches often trade off feasibility against stability and computation cost (e.g., frequent projections), rely on surrogate penalties that under-enforce equalities, or target low-dimensional regimes. In particular, none explicitly address the high-dimensional constrained optimization setting we study, where substantial equality and inequality constraints must be satisfied simultaneously.

**Reinforcement learning for optimization.** RL is extensively applied to *discrete, combinatorial optimization* (CO) (Mazyavkina et al., 2021), where the discrete nature of the search space often renders exact solvers computationally infeasible. Early work (Bello et al., 2017) shows that combining neural networks with policy gradient methods yields high-quality solutions to the traveling salesman problem (TSP), and this approach is soon extended to the vehicle routing problem (VRP) (Nazari et al., 2018). These initial successes spur a wave of research on architectural innovations, leading to consistently strong results in routing problems. A complementary line of work integrates RL with graph representation learning (Khalil et al., 2017; Chen & Tian, 2019; Lu et al., 2020; Kwon et al., 2020; Chen et al., 2021; Hottung et al., 2022; Feng

et al., 2025; Li et al., 2025c), shifting the focus from direct policy learning toward hybrid approaches that combine RL with explicit search or refinement mechanisms. By contrast, RL is only sparingly applied to *continuous optimization*. A few studies explore its use for general continuous optimization (Li & Malik, 2017a;b; Chen et al., 2022) or in domain-specific contexts (Xian et al., 2025), but none address *constrained* continuous optimization, which is the focus of our work.

## B SDE/ODE-based generative model in RL framework

### B.1 SDEs/ODEs as MDPs

A general form of a stochastic differential equation (SDE) is given by

$$dx_t = f(x_t, t)\, dt + g(x_t, t)\, dW_t, \tag{13}$$

where $f(x_t, t)$ denotes the deterministic drift, $g(x_t, t)$ specifies the diffusion coefficient, and $dW_t$ represents increments of a Brownian motion. When $g \equiv 0$, this reduces to a deterministic ordinary differential equation (ODE).

To connect with a Markov decision process (MDP), it is convenient to consider a discrete-time version with step size $\Delta t = 1$:

$$x_{t+1} = x_t + f(x_t, t) + g(x_t, t)\epsilon_t, \qquad \epsilon_t \sim \mathcal{N}(0, I). \tag{14}$$

This discretized form makes the analogy to an MDP explicit, with components of the MDP defined as follows:

- **State ($s_t$):** the system state $x_t \in \mathbb{R}^d$ together with time $t$.

- **Action ($a_t$):** the update of state is parameterized by $(f, g)$ and depends on $s_t$.

- **Transition ($p(s_{t+1} \mid s_t, a_t)$):** specified by the discretized SDE, where the next state is sampled from a Gaussian distribution given the action.

- **Reward ($r_t$):** task-dependent feedback depending on state and action.

- **Discount factor ($\gamma$):** typically $\gamma = 1$ in generative modeling, since trajectories are finite-horizon and the final output is often of primary interest.

The fundamental property of the MDP holds generally for both SDE and ODE-based dynamics: the distribution of the next state depends solely on the present state and action, independent of the past trajectory.

**Flow matching as ODE–MDP.** Flow matching can be viewed as an ODE-driven MDP with deterministic dynamics

$$dx_t = v_\theta(x_t, t)\, dt, \tag{15}$$

where $v_\theta$ is a learnable velocity field.

Training proceeds by constructing trajectories that interpolate between a start distribution $p_0$ (e.g., a Gaussian prior or initialization data) and a terminal distribution $p_1$ (the target data). Given $x_0 \sim p_0$ and $x_1 \sim p_1$, an interpolation schedule $\alpha : [0, 1] \to [0, 1]$ defines the deterministic bridge

$$\bar{x}_t = (1 - \alpha(t))\, x_0 + \alpha(t)\, x_1, \qquad t \in [0, 1], \tag{16}$$

with target velocity

$$v^\star(\bar{x}_t, t) = \dot{\alpha}(t)\, (x_1 - x_0), \tag{17}$$

where $\dot{\alpha}(t)$ denotes the derivative of the interpolation schedule and thus $v^\star$ represents the analytic ground-truth velocity along the interpolation trajectory. The flow matching objective is defined as:

$$\mathcal{L}_{\mathrm{FM}}(\theta) = \mathbb{E}_{\substack{t \sim \mathcal{U}(0,1) \\ x_0 \sim p_0,\, x_1 \sim p_1}} \left[ w(t)\, \|v_\theta(\bar{x}_t, t) - v^\star(\bar{x}_t, t)\|_2^2 \right]. \tag{18}$$

where $w(t)$ is an optional time-dependent weight.

This training process can be reinterpreted as *offline reinforcement learning* with a prescribed sampling template that generates trajectory data $(\bar{x}_t, v^\star(\bar{x}_t, t))$ connecting the start and terminal distributions along a straight line. The policy $v_\theta$ is optimized with a behavior cloning objective, imitating the analytically defined target policy $v^\star$ on these pre-collected trajectories.

Finally, one may introduce stochasticity into the dynamics:

$$dx_t = v_\theta(x_t, t)\, dt + g(t)\, dW_t, \tag{19}$$

which yields a stochastic variant of flow matching while retaining the same policy-learning interpretation.

**Diffusion models as SDE–MDP.** Song et al. (Song et al., 2021) formulate diffusion models in a unified SDE framework, specifying both forward and reverse dynamics in continuous time. The forward process gradually perturbs data into noise,

$$dx_t = f(x_t, t)\, dt + g(t)\, dW_t, \tag{20}$$

while the corresponding reverse process enables generation from noise,

$$dx_t = \left(f(x_t, t) - g^2(t)\nabla_x \log p_t(x_t)\right) dt + g(t)\, d\bar{W}_t, \tag{21}$$

where $f(\cdot)$ denotes the drift, $g(\cdot)$ the diffusion coefficient, $dW_t$ a Brownian increment, and $\bar{W}_t$ its reverse-time counterpart.

Different choices of $(f, g)$ correspond to different forward processes. A classical example is the *variance-exploding (VE) SDE*, defined by $f(x_t, t) = 0$ and $g(t) = \sqrt{\frac{d[\sigma^2(t)]}{dt}}$, where $\sigma(t)$ is an increasing variance schedule. The resulting forward marginal has the closed form $x_t = x_0 + \sigma(t)\epsilon$ with $\epsilon \sim \mathcal{N}(0, I)$, which is Gaussian with mean $x_0$ and variance $\sigma^2(t)I$. Thus, the forward process can be interpreted as sampling random RL-like trajectories that progressively transform the data distribution into a standard Gaussian distribution.

The reverse process requires access to the score function $\nabla_x \log p_t(x_t)$, which is intractable in general. In the VE case, however, the Gaussian forward marginals admit a closed form for the score, enabling efficient training of a neural network $s_\theta(x_t, t)$ to approximate it via denoising score matching:

$$\mathcal{L}_{\mathrm{DSM}}(\theta) = \mathbb{E}_{\substack{t \sim \mathcal{U}(0,1) \\ x_0 \sim p_{\mathrm{data}} \\ \epsilon \sim \mathcal{N}(0,I)}}\left[\lambda(t)\, \| s_\theta(x_t, t) + \tfrac{1}{\sigma(t)}(x_t - x_0)\|_2^2\right]. \tag{22}$$

In a similar way as we state in the previous section, this procedure can be interpreted as *offline reinforcement learning*, where the forward SDE generates trajectories and the score network $s_\theta$ is trained by behavior cloning to imitate the sampled reverse policy.

Finally, Song et al. (Song et al., 2021) show that diffusion admits a deterministic counterpart, the *probability flow ODE*, further unifying ODE- and SDE-based generative models under the same MDP perspective.

Some other works also discuss the imitation learning nature of ODE/SDE-based generative models, supporting our understanding (Ma et al., 2025; Celik et al., 2025). After discussing constrained optimization problems, ODE/SDE-based generative models in the unified framework of RL, A promising yet underexplored pathway for solving constrained optimization problems is presented. And the enriched toolbox of reinforcement learning is open.

## C   Problem Details

### C.1   Convex Quadratic Programs (QPs)

We consider convex quadratic programs of the form

$$\min_{y \in \mathbb{R}^n} \tfrac{1}{2}y^\top Q y + p^\top y, \quad \text{s.t. } Ay = x,\ Gy \leq h. \tag{23}$$

**Problem dimensions.** Following prior work, we use $n = 100, n_{\text{eq}} = 50, n_{\text{ineq}} = 50$.

**Data generation.** Matrices and vectors are sampled as in DC3 Donti et al. (2021): $Q$ is diagonal with entries sampled from $\mathcal{U}(0,1)$, $p$ also from $\mathcal{U}(0,1)$, and entries of $A$ and $G$ from $\mathcal{N}(0,1)$. For each instance, $x \sim \mathcal{U}(-1,1)^{n_{\text{eq}}}$. The inequality bound $h$ is constructed to ensure feasibility via $h_i = \sum_j |(GA^+)_{ij}|$. Ground truth solutions are generated using classical solver qpth (Stellato et al., 2020)

## C.2 Quadratic Programs with Sinusoidal Regularization (QPSR)

We evaluate mild non-convexity using the extension:

$$\min_{y \in \mathbb{R}^n} \tfrac{1}{2} y^\top Q y + p^\top \sin(y), \quad \text{s.t. } Ay = x, \ Gy \le h. \tag{24}$$

**Problem dimensions.** We use the same setting as QPs: $n = 100, n_{\text{eq}} = 50, n_{\text{ineq}} = 50$.

**Data generation.** All $(Q, p, A, G, h, x)$ are generated identically to QPs. The only difference is the sinusoidal term $p^\top \sin(y)$, which introduces structured non-convexity while keeping the constraint geometry unchanged. Ground truth solutions are generated using classical solver IPOPT (Wächter & Biegler, 2006).

## C.3 AC Optimal Power Flow (ACOPF)

We follow exactly the ACOPF formulation and experimental setup used in DC3 Donti et al. (2021). Given a power network with $b$ buses and complex voltage vector $v \in \mathbb{C}^b$, real and reactive generator outputs $(p_g, q_g) \in \mathbb{R}^b$, and demands $(p_d, q_d)$, the objective is to minimize the standard quadratic generation cost:

$$\min_{p_g, q_g, v} \quad p_g^\top A\, p_g \ + \ b^\top p_g, \tag{25}$$

subject to operational constraints:

$$p_g^{\min} \le p_g \le p_g^{\max}, \qquad q_g^{\min} \le q_g \le q_g^{\max}, \qquad v^{\min} \le |v| \le v^{\max}, \tag{26}$$

and the nonlinear AC power-flow equations:

$$(p_g - p_d) \ + \ i(q_g - q_d) \ = \ \text{diag}(v)\, W\, \overline{v}, \tag{27}$$

where $W$ is the network admittance matrix.

**Problem dimensions and generation** We use the MATPOWER case57 system with

$$|B| = 57, \quad |G| = 7, \quad |D| = 42, \quad |R| = 1,$$

where $B$ denotes all buses, $G$ generator (PV) buses, $D$ load (PQ) buses, and $R$ the slack bus. The ACOPF problem solves for

$$p_g, q_g \in \mathbb{R}^{57}, \qquad v \in \mathbb{C}^{57},$$

i.e., 57 real-power variables, 57 reactive-power variables, and 57 complex voltages (114 real degrees of freedom). Real/reactive loads, generator and voltage limits, and the admittance matrix $W$ are taken from case57, with load values perturbed multiplicatively as in DC3. Ground truth solutions are generated using PYPOWER (Zimmerman et al., 1997).

**Predicted vs. recovered variables.** Following DC3, the neural network predicts only

$$(p_g)_G, \quad |v|_{B \setminus D},$$

while all remaining variables

$$(p_g)_R, \ (q_g)_{B \setminus D}, \ |v|_D, \ (\angle v)_{B \setminus R}$$

are recovered using a two-step Newton solve, enforcing the AC power-flow equations.

Table 4: Ablation study of different penalty terms on QPs.

| Metric | l1 | l2 | l2 squared | soft barrier | augmented lagrangian |
|---|---|---|---|---|---|
| Obj. value ↓ | -14.61 (0.00) | -15.01 (0.00) | -15.11 (0.00) | 49.71 (0.00) | -15.06 (0.01) |
| Max eq viol. ↓ | 0.04 (0.00) | 0.03 (0.00) | 0.05 (0.00) | 0.05 (0.00) | 0.04 (0.00) |
| Mean eq viol. ↓ | 0.01 (0.00) | 0.01 (0.00) | 0.01 (0.00) | 0.02 (0.00) | 0.01 (0.00) |
| Eq feasisble rate ↑ | 0.00 (0.00) | 0.00 (0.00) | 0.00 (0.00) | 0.00 (0.00) | 0.00 (0.00) |
| Max ineq viol. ↓ | 0.00 (0.00) | 0.00 (0.00) | 0.03 (0.00) | 0.00 (0.00) | 0.08 (0.02) |
| Mean ineq viol. ↓ | 0.00 (0.00) | 0.00 (0.00) | 0.00 (0.00) | 0.00 (0.00) | 0.01 (0.00) |
| Ineq feasisble rate ↑ | 0.97 (0.00) | 0.82 (0.00) | 0.00 (0.00) | 1.00 (0.00) | 0.00 (0.00) |
| Feasible rate ↑ | 0.00 (0.00) | 0.00 (0.00) | 0.00 (0.00) | 0.00 (0.00) | 0.00 (0.00) |

## D   Ablation Experiments

### D.1   Ablation Study on Penalty Terms in the Loss Function

We adopt Eq. 2 as the base loss for merit optimization, where constraint violations are penalized through additional regularization terms. While our main experiments use the $\ell_2$-norm penalty, following the settings of DC3 (Donti et al., 2021), different choices of penalty formulations may lead to distinct optimization performance. In this section, we conduct an ablation study over several commonly used penalty terms for equality and inequality constraints, evaluating their relative performance within the vanilla setting.

Let $h_X(y)$ and $g_X(y)$ denote equality and inequality constraint functions, respectively, and $[\cdot]_+$ the element-wise positive part. The evaluated penalty formulations are listed below.

**$\ell_1$-norm penalty**

$$M_X^{\ell_1}(y) = f_X(y) + \alpha\|h_X(y)\|_1 + \beta\|[g_X(y)]_+\|_1. \tag{28}$$

**$\ell_2$-norm penalty**

$$M_X^{\ell_2}(y) = f_X(y) + \alpha\|h_X(y)\|_2 + \beta\|[g_X(y)]_+\|_2. \tag{29}$$

**Squared $\ell_2$-norm penalty**

$$M_X^{\ell_2^2}(y) = f_X(y) + \alpha\|h_X(y)\|_2^2 + \beta\|[g_X(y)]_+\|_2^2. \tag{30}$$

**Soft log-barrier penalty**

$$M_X^{\text{barrier}}(y) = f_X(y) + \alpha\sum_i \log\left(\text{softplus}(-|h_{X,i}(y)|)\right) + \beta\sum_j \log\left(\text{softplus}(-[g_{X,j}(y)]_+)\right). \tag{31}$$

**Augmented Lagrangian (PHR).**   We additionally evaluate the Powell–Hestenes–Rockafellar (PHR) augmented Lagrangian formulation. The loss is defined as

$$M_X^{\text{AL}}(y) = f_X(y) + \lambda_h h_X(y) + \frac{\rho}{2}\|h_X(y)\|_2^2 + \frac{\rho}{2}\left(\left\|\left[\frac{\lambda_g}{\rho} + g_X(y)\right]_+\right\|_2^2 - \left\|\frac{\lambda_g}{\rho}\right\|_2^2\right), \tag{32}$$

where $\lambda_h$ and $\lambda_g$ denote the Lagrange multipliers associated with equality and inequality constraints, respectively, and $\rho > 0$ are penalty parameters. In our implementation, the multiplier, including $\lambda_h$ and $\lambda_g$, updates are performed outside the network optimization loop, while Eq. equation 32 is used as a training loss to update model parameters. Due to the known difficulty of optimizing augmented Lagrangian objectives in end-to-end learning settings, we extend the training horizon to 10,000 epochs for this variant.

As shown in Table 4, none of the evaluated penalty-based loss formulations can produce solutions that satisfy the equality constraints, indicating that soft constraint penalties alone are insufficient for enforcing strict

feasibility without additional mechanisms. Among the tested formulations, $\ell_1$, $\ell_2$, squared $\ell_2$, and augmented Lagrangian losses achieve comparable objective values, suggesting that these penalties are generally effective in guiding optimization. With respect to inequality constraints, $\ell_1$, $\ell_2$, and soft barrier penalties demonstrate relatively strong feasibility performance, achieving high feasible rates for inequality constraints.

While the PHR augmented Lagrangian provides a theoretically principled framework for constrained optimization, our empirical results indicate that treating it as a standard training objective, even with Lagrange multipliers updated via a discrete outer-loop procedure, is often insufficient to drive constraint residuals to zero. This difficulty arises from the complex interplay between the stochastic nature of neural network optimization and the non-stationary loss landscape induced by the updates of multipliers. Unlike classical optimization, where primal subproblems are solved to high precision (Rockafellar, 1974), the inexact and non-convex minimization inherent in neural network training hinders the systematic elimination of violations. Our findings align with recent perspectives (Kotary & Fioretto, 2024; Boero et al., 2025) that suggest the Augmented Lagrangian should be viewed as a procedural algorithm rather than a loss function; when the underlying algorithmic structure and convergence requirements are decoupled from the training loop, the theoretical guarantees of the PHR formulation no longer strictly hold.

### D.2 Ablation Study on Penalty Weight in Loss

In our primary experiments, the penalty weight is set to 10, following the convention adopted by DC3 (Donti et al., 2021). To evaluate the effect of this hyperparameter, we conduct an ablation study on the validation set of quadratic programming (QP) problems, while keeping all other experimental settings fixed.

As shown in Figure 5, increasing the penalty weight leads to higher feasibility rates but also larger objective values (lower is better), reflecting a trade-off between constraint satisfaction and objective optimality. In particular, feasibility improves rapidly as the penalty weight increases from 4 to 10, after which the gains become marginal. This observation motivates our choice of 10 as a reasonable balance point for the penalty parameter.

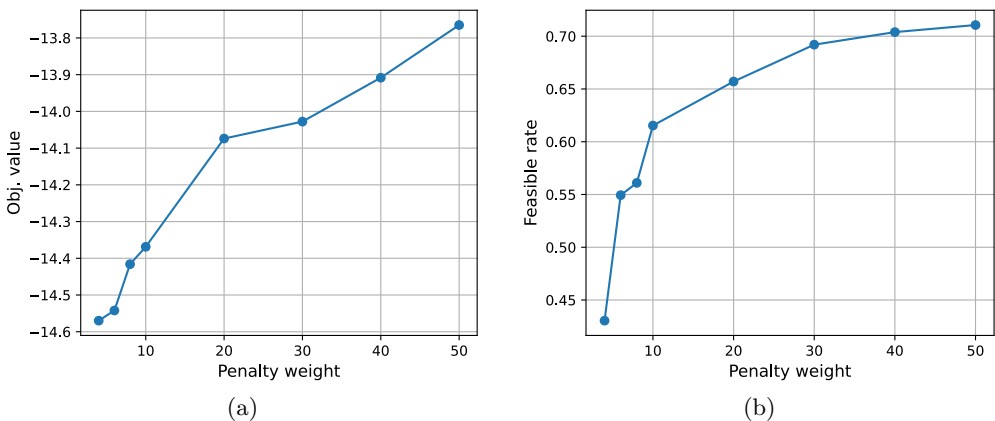

(a)                    (b)

Figure 5: Objective value (lower is better) (a) and feasible rate (b) under different penalty weights on QPs.

### D.3 Ablation Study on cutoff for *RelaxComp*

In this section, we investigate the sensitivity of RELAXCOMPLETION (Algorithm 2) to the cutoff parameter $c$, which limits how significantly the clipping operation can change the initial solution $y$ based on the completion residual. We conduct this ablation study on the validation set of quadratic programming (QP) problems, while keeping all other experimental settings fixed.

As shown in Figure 6b, the feasible rate is nearly zero for $c \leq 0.2$, indicating that overly restrictive clipping severely limits the completion operator's ability to drive infeasible solutions toward the feasible region; in this regime, the method effectively reduces to the Vanilla CMFO variant. A sharp transition is observed at

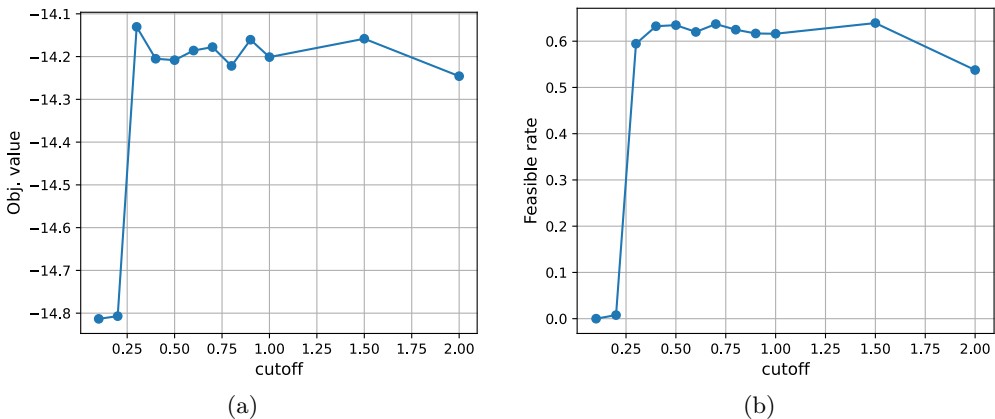

Figure 6: Objective value (lower is better) (a) and feasible rate (b) under different cutoffs on QPs.

$c = 0.3$, where the feasible rate abruptly increases to above 0.6, after which it remains consistently high with only minor oscillations across a wide range of cutoff values. For very large cutoffs (e.g., $c \in [1.5, 2.0]$), the feasible rate exhibits a slight decline, reflecting the increasing similarity of the algorithm's behavior to the unconstrained Completion variant as the clipping constraint becomes loose. Figure 6a shows a corresponding pattern in the objective value (lower is better). While very small cutoffs yield lower objective values, this occurs in a regime where feasibility is rarely achieved. A sharp transition is also observed around $c = 0.3$, after which the objective value enters a broad plateau and remains relatively stable across a wide range of cutoff values.

Together, these results highlight a clear but mild trade-off between constraint satisfaction and objective optimization, and suggest that the algorithm is relatively insensitive to the choice of $c$ over a broad intermediate range. Based on this empirical robustness, we select $c = 0.5$ for the main experiments, as it achieves a good balance between a high feasible rate and competitive objective values. For simplicity, we fix this hyperparameter across all datasets, even though the ablation is conducted only on QPs. Dataset-specific tuning of $c$ may further improve performance.

## E    Limitation

While CMFO achieves strong empirical performance on the evaluated benchmarks, several limitations should be noted. First, the current experimental evaluation is limited to two synthetic datasets (QP and QPSR) and one real-world power grid dataset (ACOPF), where the solution dimensionality does not exceed 200. The scalability and effectiveness of CMFO in substantially higher-dimensional settings remain unverified. Second, CMFO adopts a soft constraint enforcement strategy based on penalty terms and RELAXCOMPLETION. In high-stakes applications where strict or exact feasibility is required, such soft enforcement alone may be insufficient, and additional post-processing or constraint-handling mechanisms may still be necessary. Finally, we would like to point out that while the proposed framework is general, separate models are trained for different problem families, and the method is not intended to generalize across problem families with a single trained model.