# OpenReview forum: "Solving Constrained Optimization Problems as ODE-based Models Using Reinforcement Learning"
_TMLR — Accepted by TMLR_

### Review · Reviewer_hnVM · 2025-12-10

**Summary Of Contributions:**

## Summary
This paper presents a novel framework for solving constrained optimization problems by integrating Reinforcement Learning (RL) with ODE-based generative models, particularly flow matching. It reformulates constrained optimization as a Markov Decision Process (MDP), thereby viewing the iterative refinement of candidate solutions as a sequential decision-making process. Building on this perspective, the work establishes an equivalence between RL and ODE-based generative models, unifying them under a common iterative-dynamic interpretation as trainable optimizers. At the core of the proposed method is the Constrained Markov Flow Optimizer (CMFO), which combines flow matching with a soft constraint-enforcement mechanism named relaxCompletion—designed to preserve gradient signals while steering solutions toward feasibility. The framework is evaluated on a range of convex, non-convex, and real-world optimization tasks, demonstrating competitive performance in terms of solution quality, constraint satisfaction, and inference speed compared to existing learning-based and classical solvers.

### Key Strengths

The MDP-based reformulation is intuitive and well-motivated. The relaxCompletion operator is a practical and thoughtful solution to the challenge of enforcing equality constraints without losing training signals. Experiments are convincing and cover a range of problem types, with clear ablation studies.

### Key Weaknesses

1. The paper reformulates the constrained problem (Eq. 1) into an unconstrained merit function (Eq. 2) for the MDP reward. This introduces some approximation gap: minimizing the merit function is not rigorously equivalent to solving the original constrained problem, unless the penalty weights $(\alpha, \beta)$ go to infinity. The paper does not theoretically analyze this gap or discuss how it might affect the final solution's feasibility and optimality. Empirically, the choice $\alpha=\beta=5$ is stated but not justified or ablated, leaving the robustness of this critical design choice unclear.

2. While strong baselines are included, the evaluation does not compare against several recently published diffusion-for-optimization methods (e.g., Diffusolve, Gauge Flow Matching), which are relevant and would better situate the performance claims.

**Audience:**

Yes

**Audience Explanation:**

TMLR’s audience includes researchers in machine learning, optimization, control, and applied mathematics. This work sits at the intersection of several active research areas: learning-to-optimize (L2O), diffusion/flow-based generative models, and reinforcement learning. The proposed framework offers a fresh perspective on how to leverage generative models for constrained optimization: a topic of growing interest in both theoretical and applied communities. Readers in fields such as power systems, robotics, and logistics could also benefit from the efficiency gains demonstrated in the experiments.

**Broader Impact Concerns:**

I don't see any missing concerns on the ethical implications need to be addressed in the paper.

**Claims And Evidence:**

Yes

**Claims Explanation:**

The paper links constrained optimization to MDPs and ODE-based generative models. The experimental section is convincing, with results reported across three distinct tasks (QP, QPSR, ACOPF). Comparisons with other classical methods (Vanilla, Completion, +RelaxComp) demonstrate the contribution of each component. Tables and figures are informative and support the claims regarding trade-offs between optimality, feasibility, and speed.

**Requested Changes:**

The major concerns or weaknesses have been stated in the Summary: Key weaknesses. In addition, I have the following minor suggestions:

1. Please consider to make the best values in the tables bold, to clearly show the best method in different tasks.
2. Please consider to briefly discuss settings where CMFO might struggle (e.g., highly discontinuous constraints, very high-dimensional problems) and potential extensions.
3. There is a clipping hyper-parameter $c$ introduced in Algorithm 2. $c$ is set to be $0.5$ in the experiments. How should $c$ be chosen in the experiments? There seems to be a trade-off between the performance and feasibility.

---

> ### Author Response · Authors · 2026-01-10
>
> Thank you for raising these thoughtful and constructive questions. We address each point in turn below.
>
> > **1. Minimizing the merit function is not rigorously equivalent to solving the original constrained problem, unless the penalty weights go to infinity.**
>
> We agree that reformulating the constrained problem in Eq. (1) using a finite-penalty merit function in Eq. (2) introduces an approximation gap and is not theoretically equivalent to the original constrained formulation unless the penalty weights tend to infinity. Our goal is not to claim strict equivalence, but to adopt a practical and widely used surrogate that is amenable to learning-based optimization and reinforcement learning formulations.
>
> To empirically justify this design choice, we provide a sensitivity analysis of the penalty weight in Appendix D.2. The results demonstrate a clear trade-off between objective value and feasibility as the penalty weight increases. This analysis supports our choice of setting the penalty weight to 10, which achieves a favorable balance between constraint satisfaction and objective quality in practice.
>
> > **2. Baseline selection.**
>
> We include DC3 (Donti et al., 2021) and B-Projection (Liang & Chen, 2025) as representative state-of-the-art learning-based baselines for constrained optimization under our problem setting, which emphasizes fast inference, near-optimal objective values, and extremely tight constraint tolerances (violation < 1e−4).
>
> Below we clarify our rationale for excluding the other cited methods:
>
> 1. Gauge Flow Matching (Li et al., 2025b) primarily focuses on enforcing feasibility through gauge mappings, while largely de-emphasizing objective optimality. This design choice does not align with our setting, where both feasibility and objective quality are critical. Moreover, B-Projection (Liang & Chen, 2025), which is developed by the same group and included as one of our baselines, discusses Gauge Flow Matching and provides a stronger and more relevant comparison for our target regime.
>
> 2. DiffuSolve (Li et al., 2025a) is designed to predict initializations for downstream optimizers, and its performance is therefore tightly coupled to the specific solver employed, which may vary across tasks. The paper primarily targets trajectory prediction benchmarks, relies on task-specific optimization procedures, and does not release code, making fair and reproducible comparison infeasible in our setting.
>
> Given these considerations, we believe that the selected baselines provide a fair and representative comparison for our target setting.
>
> > **3. Please consider to make the best values in the tables bold, to clearly show the best method in different tasks.**
>
> We agree and have updated the tables accordingly to clearly highlight the best-performing methods for metrics (Obj. value, Feasible rate, and Time (s)). It should be noted that a good method should achieve good performance on all three metrics. Achieving the best performance on only one metrics is usually meaningless.
>
> > **4. Please consider to briefly discuss settings where CMFO might struggle and potential extensions.**
>
> We have added a discussion of the limitations of CMFO in Appendix E. In the current study, CMFO is evaluated on two synthetic datasets (QP and QPSR) and one real-world power grid dataset (ACOPF), where the solution dimensionality does not exceed 200. As a result, the scalability and effectiveness of CMFO in substantially higher-dimensional settings remain unverified. In addition, CMFO adopts a soft constraint enforcement strategy based on penalty terms and \textsc{relaxCompletion}, which may be insufficient for high-stakes applications that require strict or exact feasibility guarantees.
>
> > **5. How should $c$ be chosen in the experiments?**
>
> We have added an ablation study on the cutoff parameter $c$ in Appendix D.3 to clarify its impact on performance and provide guidance for its selection. The results demonstrate a clear trade-off between feasibility and objective quality: while increasing $c$ in a range improves the feasible rate, it leads to a degradation in the objective value. Empirically, we observe a sharp performance transition around $c=0.3$, followed by a broad plateau where value remain relatively stable for both feasibility and objective. This indicates that the algorithm is robust to the choice of $c$ within this operative range. Based on this observed balance, we selected $c=0.5$ as a robust default for all experiments.

---

### Review · Reviewer_UoDQ · 2025-12-23

**Summary Of Contributions:**

## Summary:
The paper proposes a learning-to-optimize method that formulates optimization steps as denoising steps in a diffusion model framework. The authors present an interesting unification of optimization under reinforcement learning principles. Additionally, the paper introduces a novel method to address training instability arising from equality constraints.

## Strengths:
- The unification of Markov decision process, flow match and Q-learning is inspiring.
- RelaxCompletion on Equality Constraints is very smart.
- The proposed method has been validated on multiple datasets and showed the improvement in time efficiency while having reasonable convergence.

## Weakness:

The paper has several weaknesses as below.

- I am surprised by the finding that the vanilla method completely fails to achieve equality feasibility (0% across all benchmarks). The paper attributes this to L1 penalty limitations, but this explanation lacks depth. I recommend: a detailed ablation study examining different penalty formulations (L1, L2, log-barrier, augmented Lagrangian),  analysis of gradient behavior near constraint boundaries and the discussion of why standard penalty methods fail in the flow-matching context. This analysis would strengthen the motivation for RelaxCompletion and make the contribution more convincing.

- The paper does not discuss whether starting from solutions sampled from the equality-feasible set is possible or beneficial. For problems where feasible sampling is tractable (e.g., linear equality constraints), this could provide a stronger baseline and potentially improve performance.

- The paper lacks analysis of the optimizer's generalization capability. I’m concerned that the learned optimizer overfits the training distribution. I recommend adding experiments with distribution shift or cross-validation across problem families to demonstrate robustness.

- How do you collect the computation time of different methods? It’s important to be clarified.  I presume your method would be run on GPU whereas the traditional optimizer might be run on CPU. So it should be reported to be clear.

- I’m probably missing something. The baselines seem insufficient to me. Because the paper lists some papers on learning to optimize in the related work section. But they’re not compared in the experiments.

- The paper focuses on final solution quality and computation time but does not analyze convergence rates.

**Audience:**

Yes

**Audience Explanation:**

This is a novel theoretical paper, which many folks would find it interesting.

**Claims And Evidence:**

No

**Claims Explanation:**

The computation comparison is not convincinving without clarifying how the computation time is collected.
Furthermore, the claim can be better supported with more experimental analysis such as generalization capability analysis and convergence rate etc, as I detailed in the weaknesses section above.

**Requested Changes:**

See my recommendation in the weaknesses section where I explain why it's weak and how to improve them.

---

> ### Author Response · Authors · 2026-01-10
>
> We appreciate the reviewer’s careful reading and thoughtful feedback. We respond to each comment below.
>
> > **1. Penalty formulations in loss function.**
>
> We have added a systematic ablation study in Appendix D.1 that compares multiple commonly used penalty formulations, including $\ell_1$, $\ell_2$, squared $\ell_2$, soft log-barrier, and augmented Lagrangian penalties, under the same flow-matching optimization setting.
> Empirically, the results show that none of the evaluated penalty-based formulations are able to enforce equality feasibility in the vanilla flow-matching setting, even when penalty weights are increased or when augmented Lagrangian multipliers are updated via an outer-loop procedure. While these penalties are effective for guiding objective optimization and, in some cases, enforcing inequality constraints, equality violations persist across all variants.
> These findings directly motivate the design of RelaxCompletion, which enforces equality feasibility through a controlled corrective mechanism while preserving informative learning signals. Further analysis and detailed results are provided in Appendix D.1.
>
> > **2. Sampling from the equality-feasible set.**
>
> We agree that sampling from the equality-feasible set can be beneficial when such sampling is tractable. This setting is already covered by the Completion baseline, which explicitly enforces equality feasibility by predicting solutions in a reduced variable space and completing them via equality constraints (Section 4.2). As shown in the main results, this baseline substantially improves equality feasibility compared to vanilla flow matching.
> Building on this approach, RelaxCompletion further improves both feasibility and objective quality by introducing controlled, gradient-preserving corrections. Empirically, RelaxCompletion consistently outperforms the Completion baseline across all evaluated benchmarks, demonstrating the superior of our method.
>
> > **3. Optimizer's generalization capability.**
>
> Thank you for raising this important concern. We clarify that learning-to-optimize methods, including CMFO, are developed under the assumption that instances within the same problem family share common structural properties, such as constraint forms, variable semantics, and underlying KKT geometry. Accordingly, separate models are trained for different problem families (e.g., QP, QPSR, ACOPF), which is standard practice in the learning-to-optimize literature [1,2].
> Under this setting, generalization is evaluated across unseen instances drawn from the same problem family rather than across fundamentally different optimization families. Cross-family generalization or large distribution shifts would correspond to transferring between distinct constrained optimization problems with different structures, which is outside the scope of the current formulation.
>
> [1]Priya Donti, David Rolnick, and J Zico Kolter. Dc3: A learning method for optimization with hard constraints. In International Conference on Learning Representations, 2021.
>
> [2]Tianlong Chen, Xiaohan Chen, Wuyang Chen, Howard Heaton, Jialin Liu, Zhangyang Wang, and Wotao Yin. Learning to optimize: A primer and a benchmark. Journal of Machine Learning Research, 23(189): 1–59, 2022.
>
> > **4. Computation time collection of different methods?**
>
> All classical optimizers are implemented following the DC3 codebase and executed entirely on GPU (A40), using the same hardware configuration as the learning-based methods. All reported runtimes are measured under identical conditions.

---

> ### Author Response · Authors · 2026-01-10
>
> > **5. Baseline selection.**
>
> We include DC3 (Donti et al., 2021) and B-Projection (Liang & Chen, 2025) as representative state-of-the-art learning-based baselines for constrained optimization under our problem setting, which emphasizes fast inference, near-optimal objective values, and extremely tight constraint tolerances (violation < 1e−4).
>
> Below we clarify our rationale for excluding the other cited methods:
>
> 1. Gauge Flow Matching (Li et al., 2025b) primarily focuses on enforcing feasibility through gauge mappings, while largely de-emphasizing objective optimality. This design choice does not align with our setting, where both feasibility and objective quality are critical. Moreover, B-Projection (Liang & Chen, 2025), which is developed by the same group and included as one of our baselines, discusses Gauge Flow Matching and provides a stronger and more relevant comparison for our target regime.
>
> 2. DiffuSolve (Li et al., 2025a) is designed to predict initializations for downstream optimizers, and its performance is therefore tightly coupled to the specific solver employed, which may vary across tasks. The paper primarily targets trajectory prediction benchmarks, relies on task-specific optimization procedures, and does not release code, making fair and reproducible comparison infeasible in our setting.
>
> 3. MBD (Pan et al., 2024): when adapted to our problem formulation, this approach becomes equivalent to a diffusion-based variant of the VANILLA method, relying solely on penalty terms in the loss function to encourage feasibility. Diffusion-based methods rely on stochastic sampling trajectories, which introduce variance in constraint satisfaction and make it difficult to strictly enforce constraints at tight tolerances. As a result, they are not well aligned with our setting, which emphasizes a strict feasibility tolerance (1e−4). As shown in Appendix D.1, where we evaluate different penalty terms for VANILLA, relying solely on penalty terms in the loss function totally fails to satisfy equality constraints. Based on this empirical evidence, we do not expect MBD methods to produce any feasible solutions under our constraints, and therefore do not include them as baselines.
>
> Given these considerations, we believe that the selected baselines provide a fair and representative comparison for our target setting.
>
> > **6. The paper focuses on final solution quality and computation time but does not analyze convergence rates.**
>
> Thank you for raising this point. Our evaluation focuses on final solution quality and inference time because these metrics are the most relevant for the downstream applications we target, where the goal is to obtain a high-quality feasible solution under real-time constraints. As discussed in the paper, the flow-matching refinement process is trained end-to-end to optimize terminal performance. Accordingly, we focus our analysis on the quality of the final solution produced, rather than on intermediate convergence behavior.

---

> > ### Comment · Reviewer_UoDQ · 2026-02-09
> >
> > Thanks for the clarification!
> >
> > I think my concerns have been addressed mostly and I will leave the final decision to the AC.  Regardless the final decision, I would like to request authors make it clear in limitation section that the method is not able to generalize into different problem families. Because this is a general questions that readers usually have.

---

> > > ### Author Response · Authors · 2026-02-09
> > >
> > > Thank you very much for your thoughtful comment!
> > >
> > > We will add a clarification in the Limitations section to make it explicit that, while the proposed framework is general, separate models are trained for different problem families, and the method is not intended to generalize across problem families with a single trained model. We believe this clarification will help avoid potential confusion for readers.

---

### Review · Reviewer_MQoH · 2026-01-04

**Summary Of Contributions:**

The paper proposes the Constrained Markov Flow Optimizer (CMFO), a framework that unifies Reinforcement Learning (RL) and Ordinary Differential Equation (ODE)-based generative models (specifically Flow Matching) to solve constrained optimization problems . The authors formalize constrained optimization as a Markov Decision Process (MDP) where the state is the candidate solution, and the action is the refinement step .

The primary technical contribution is the introduction of the `relaxCompletion` (relaxComp) operator. Unlike previous methods that strictly project solutions onto the equality manifold (often blocking gradient signals), `relaxCompletion` applies a clipped residual correction. This allows the method to guide solutions toward feasibility while preserving the gradient information necessary for training the underlying flow-matching policy .

The method is evaluated on convex Quadratic Programs (QPs), non-convex QPs with sine regularization (QPSR), and AC Optimal Power Flow (ACOPF) tasks . The authors claim that CMFO achieves a better trade-off between solution optimality, feasibility, and inference speed compared to baselines such as DC3 and vanilla flow-matching implementations .

**Audience:**

Yes

**Audience Explanation:**

The paper addresses "Learning to Optimize" (L2O), a significant and active area of research within the machine learning community . The work sits at the intersection of three popular domains:

1. **Generative Modeling:** Specifically Flow Matching and ODE-based models .
2. **Reinforcement Learning:** Formulating optimization as a sequential decision process .
3. **Constrained Optimization:** With practical applications in power systems and robotics .

TMLR readers interested in the theoretical unification of RL and generative models, as well as practitioners looking for neural solvers for complex engineering problems like ACOPF, would find the proposed framework and the `relaxCompletion` operator relevant.

**Broader Impact Concerns:**

The authors address optimization problems common in engineering (e.g., Power Flow). While the application domain (critical infrastructure) carries inherent risks if invalid solutions are deployed, the paper presents a methodological contribution rather than a deployed system. The standard Broader Impact considerations regarding energy consumption of training deep learning models apply, but no specific ethical concerns regarding bias or misuse are immediately apparent in this work. The paper adequately focuses on technical improvements in solver efficiency.

**Claims And Evidence:**

Yes

**Claims Explanation:**

The submission provides a coherent theoretical derivation linking the iterative refinement of optimization solutions to MDPs and Flow Matching dynamics . The empirical evidence generally supports the claims of improvement over the selected baselines (DC3 and B-Projection). Specifically:

* **Ablation Studies:** The authors provide clear evidence regarding the necessity of their core contribution, the `relaxCompletion` operator. Tables 1, 2, and 3 distinctly show that the "Vanilla" flow matching method fails to satisfy equality constraints (0% feasibility), while the "Completion" baseline improves feasibility but degrades optimality. The proposed method strikes a balance between the two .
* **Performance Metrics:** The paper uses standard metrics (Objective Value, Feasibility Rate, Violation Mean/Max) to quantify performance . The results in the tables indicate that CMFO achieves superior objective values compared to DC3 while maintaining competitive inference speeds .

However, the "convincing" nature of the evidence is slightly diminished by the choice of baselines. The paper cites several very recent diffusion-based constrained optimization methods from 2024 and 2025 (e.g., Li et al., 2025b; Pan et al., 2024) in the Related Work section , but does not compare against them in the experimental section, relying instead on the older DC3 (2021) and B-Projection . While the provided evidence supports the claim that CMFO is better than DC3, it is less clear if it is superior to concurrent state-of-the-art generative approaches.

**Requested Changes:**

1. **Clarification of the "RL/Q-Learning" Formulation:**
The paper frames the method heavily around Reinforcement Learning, defining a "Q-learning loss" () in Equation 11 and Algorithm 1 . However, the description suggests that this is essentially a supervised loss minimizing the merit function at the terminal state or along the trajectory, rather than learning a value function  to approximate future returns via Bellman updates.
* **Request:** Please clarify the implementation of . Is this true Q-learning with a target network and Bellman error, or is it a differentiable loss applied to the output of the ODE solver? If it is the latter, the terminology "Q-learning" may be misleading and should be rephrased to reflect that it is a merit-function minimization via differentiable unrolling or direct policy optimization.


2. **Feasibility Analysis for Real-World Tasks:**
In the ACOPF task (Table 3), the proposed method achieves a Feasibility Rate of only 54% , whereas the classical solver achieves 100% (implied, as it is a hard solver).
* **Request:** The authors must discuss the practical implications of a 54% feasibility rate. In safety-critical systems like power grids, valid solutions are required. Please add a discussion or experiment showing whether these infeasible solutions can be easily projected to the feasible set using a post-processing step, and how that affects the total "Time (s)" reported.


3. **Comparison with Cited SOTA:**
The introduction and related work mention recent diffusion-based methods for constrained optimization, specifically *Li et al. (2025a, 2025b)* and *Pan et al. (2024)* .
* **Request:** Since the paper claims improvements in "inference speed, solution quality, and feasibility over prior baselines" , it is critical to address why these recent relevant methods were not included in the quantitative comparison. If direct comparison is not feasible, a qualitative discussion explaining the theoretical advantages of CMFO over these specific recent works is necessary to contextualize the contribution.



**Minor Changes**

1. **Sensitivity Analysis of the Cutoff Parameter:**
The `relaxCompletion` operator relies on a cutoff parameter , set to  in experiments .
* **Request:** Include a sensitivity analysis (e.g., a plot or table in the Appendix) showing how performance (optimality vs. feasibility) varies as  changes. This would validate the robustness of the method and justify the choice of .


2. **Computational Overhead of Completion:**
The method uses an equality completion operator , which may involve implicit mappings or Newton's method .
* **Request:** Please provide a breakdown of the inference time. How much of the reported time is spent in the neural network inference versus the completion/correction steps? This is important to support the claim that the method is suitable for "real-time scenarios" .

---

> ### Author Response · Authors · 2026-01-10
>
> We thank the reviewer for the detailed and constructive feedback! We address each comment in turn below.
>
> >**1. Clarification of the "RL/Q-Learning" Formulation**
>
> We agree with the reviewer that our implementation should be more precisely described as policy optimization rather than Q-learning. As discussed in Section 3.2, in our setting, maximizing the return of a rollout is equivalent to minimizing the terminal metric function. Consequently, our method can be interpreted as optimizing a policy by maximizing the expected accumulated return (i.e., the Q-value), rather than performing temporal-difference–based Q-learning.
>
> To avoid any potential confusion or misinterpretation, we have revised the terminology throughout the paper accordingly.
>
> >**2. Feasibility Analysis for Real-World Tasks**
>
> A feasibility rate of 54% indicates that our method directly produces feasible solutions for more than half of the evaluated tasks. For the remaining cases, the predictions generated by the CMFO variant VANILLA, that is, the solution prior to applying the RelaxComp completion step, can be used as high-quality initializations for conventional iterative solvers.
>
> As shown in Table 3 and Figure 2(b), VANILLA consistently predicts solutions that are close to the optimum while already satisfying the inequality constraints. This enables downstream solvers to refine feasibility with fewer iterations when used as a warm start. In terms of efficiency, VANILLA requires approximately 0.003 seconds per prediction, compared to about 0.514 seconds for running an optimizer from scratch, making the cost of generating a warm-start initialization negligible relative to full optimization.
>
> Therefore, even when strict feasibility is not achieved directly, our approach can substantially accelerate the overall optimization pipeline while preserving competitive objective values.
>
> >**3. Comparison with Cited SOTA: The introduction and related work mention recent diffusion-based methods for constrained optimization, specifically Li et al. (2025a, 2025b) and Pan et al. (2024).**
>
> We include DC3 (Donti et al., 2021) and B-Projection (Liang & Chen, 2025) as representative state-of-the-art learning-based baselines for constrained optimization under our problem setting, which emphasizes fast inference, near-optimal objective values, and extremely tight constraint tolerances (violation < 1e−4).
>
> Below we clarify our rationale for excluding the other cited methods:
>
> 1. Gauge Flow Matching (Li et al., 2025b) primarily focuses on enforcing feasibility through gauge mappings, while largely de-emphasizing objective optimality. This design choice does not align with our setting, where both feasibility and objective quality are critical. Moreover, B-Projection (Liang & Chen, 2025), which is developed by the same group and included as one of our baselines, discusses Gauge Flow Matching and provides a stronger and more relevant comparison for our target regime.
>
> 2. DiffuSolve (Li et al., 2025a) is designed to predict initializations for downstream optimizers, and its performance is therefore tightly coupled to the specific solver employed, which may vary across tasks. The paper primarily targets trajectory prediction benchmarks, relies on task-specific optimization procedures, and does not release code, making fair and reproducible comparison infeasible in our setting.
>
> 3. MBD (Pan et al., 2024): when adapted to our problem formulation, this approach becomes equivalent to a diffusion-based variant of the VANILLA method, relying solely on penalty terms in the loss function to encourage feasibility. Diffusion-based methods rely on stochastic sampling trajectories, which introduce variance in constraint satisfaction and make it difficult to strictly enforce constraints at tight tolerances. As a result, they are not well aligned with our setting, which emphasizes a strict feasibility tolerance (1e−4). As shown in Appendix D.1, where we evaluate different penalty terms for VANILLA, relying solely on penalty terms in the loss function totally fails to satisfy equality constraints. Based on this empirical evidence, we do not expect MBD methods to produce any feasible solutions under our constraints, and therefore do not include them as baselines.
>
> Given these considerations, we believe that the selected baselines provide a fair and representative comparison for our target setting.
>
> >**4. Sensitivity Analysis of the Cutoff Parameter**
>
> We have included a detailed sensitivity analysis of the cutoff parameter in Appendix D.3.

---

> ### Author Response · Authors · 2026-01-10
>
> >**5. Computational Overhead of Completion**
>
> The computational overhead introduced by the completion step can be quantified by comparing inference times between CMFO-VANILLA and CMFO + RelaxComp. CMFO-VANILLA consists solely of neural network inference, whereas CMFO + RelaxComp adds a completion/correction stage.
>
> Empirically, neural network inference requires approximately 0.002–0.003 seconds per instance across all tasks. The RelaxComp step incurs an additional cost of about 0.006 seconds for QP and QPSR tasks, and approximately 0.038 seconds for ACOPF. This overhead is modest relative to the gains in feasibility and solution quality.

---

### Comment · Action_Editor_LgDm · 2026-03-06

Dear Reviewer MQoH,

The deadline for responding to the authors' rebuttal has passed, and we are still missing your feedback.

Please read the authors' response, acknowledge it in the discussion thread, and update your rating if necessary as soon as possible. We cannot move forward with the final decision without your input.

Best,
Action Editors

---

### Author Response · Authors · 2026-03-24
**Summary of Rebuttal**

Dear action editor and reviewers,

We hope this message finds you well! We are writing to provide a summary of the rebuttal period for our submission.

All three reviewers (MQoH, UoDQ, and hnVM) acknowledged that the findings of this paper are of interest to the TMLR community. Two reviewers (MQoH, hnVM) confirmed that the claims are supported by accurate, convincing, and clear evidence.

Reviewer UoDQ raised concerns regarding computation time reporting and the problem setting. During the rebuttal, we clarified the computation time methodology, and Reviewer UoDQ acknowledged that their concerns had been mostly addressed, requesting only that we strengthen the limitations section with an explicit note on problem-family generalization. We have updated the paper accordingly.

We believe all raised concerns have been addressed. Please let us know if there are any further questions or clarifications needed.

We sincerely thank you for your time and effort in handling our submission.

Best regards,
TMLR Paper6760 Authors

---

### Author Response · Authors · 2026-04-29
**Polite Follow-up on Submission Status**

Dear action editor and reviewers,

We hope this message finds you well! We are writing to respectfully follow up on the status of our submission.

As noted in our previous summary, all three reviewers acknowledged the relevance of our work to the TMLR community, and Reviewer UoDQ confirmed that their concerns had been mostly addressed; we have also incorporated the requested clarification on problem-family generalization into the Limitations section.

We fully understand that reviewing takes time and that you are managing many responsibilities. We just wanted to gently check in and ask whether there is any additional information or clarification we could provide that would help move the discussion toward a decision.

Thank you very much for your time and effort in handling our submission.

Best regards,
TMLR Paper6760 Authors

---

### Decision · Action_Editor_LgDm · 2026-05-24

**Recommendation:** Accept with minor revision

**Audience:**

Yes

**Audience Explanation:**

The paper addresses topics of interest to the TMLR community, including learning-to-optimize, generative modeling, reinforcement learning, and constrained optimization. The proposed formulation and optimization framework may be of interest to researchers working on optimization-oriented generative models and related applications.

**Claims And Evidence:**

Yes

**Claims Explanation:**

The paper presents a technically sound framework that connects constrained optimization, reinforcement learning, and ODE-based generative models. The empirical evaluation across convex, non-convex, and real-world optimization tasks generally supports the main claims regarding feasibility, solution quality, and inference efficiency. The rebuttal and revisions adequately addressed the reviewers’ concerns regarding terminology, runtime reporting, baseline selection rationale, and the discussion of generalization limitations.